# Maladaptive Compensatory Neural Mechanisms Associated with Activity-Related Osteoarthritis Pain: Dissociation of Psychological and Activity-Related Neural Mechanisms of WOMAC Pain and VAS Pain

**DOI:** 10.3390/jcm14113633

**Published:** 2025-05-22

**Authors:** Marta Imamura, Kevin Pacheco-Barrios, Paulo S. de Melo, Anna Marduy, Linamara Battistella, Felipe Fregni

**Affiliations:** 1Instituto de Medicina Fisica e Reabilitacao, IMREA, Hospital das Clinicas HCFMUSP, Faculdade de Medicina, Universidade de São Paulo, Sao Paulo 01246-000, Brazil; marta.imamura@fm.usp.br (M.I.); linamara.battistella@hc.fm.usp.br (L.B.); 2Neuromodulation Center and Center for Clinical Research Learning, Spaulding Rehabilitation Hospital and Massachusetts General Hospital, Harvard Medical School, Boston, MA 02138, USA; kpachecobarrios@mgh.harvard.edu (K.P.-B.); melosrpaulo@gmail.com (P.S.d.M.); anna.marduy@gmail.com (A.M.); 3Unidad de Investigación para la Generación y Síntesis de Evidencias en Salud, Vicerrectorado de Investigación, Universidad San Ignacio de Loyola, Lima 15023, Peru

**Keywords:** chronic pain, biomarkers, electroencephalography, osteoarthritis, rehabilitation

## Abstract

**Background/Objectives:** Knee osteoarthritis (KOA) is one of the most common causes of chronic pain and disability in older adults. Its mechanisms are both peripheral and central, causing discordance between pain intensity and disease severity. To provide better, mechanism-driven treatments for KOA, it is important to understand the emotional, physical, and neurophysiological factors that influence pain intensity. Thus, we proposed a multivariate model investigation of the multimodal predictors of pain intensity in patients with chronic KOA pain. **Methods**: We conducted an extensive assessment of 105 KOA patients. We used two different types of outcomes: (i) activity-related (Western Ontario and McMaster Universities Osteoarthritis [WOMAC] pain scale), and (ii) non-specific (visual analog scale [VAS]) pain assessments. **Results:** We found the following. (1) A higher WOMAC pain score was predicted by sensory–motor markers (lower intracortical inhibition [*p* = 0.021] and higher beta-band oscillations [*p* = 0.027]) and central sensitization (dysfunctional CPM response [*p* < 0.001]), in addition to the psychological and peripheral sensitization factors (adjusted R^2^ = 52%, F (5, 99) = 22.81, *p* < 0.0001). (2) Conversely, higher VAS pain intensity was only predicted by psychological factors (higher depression [*p* = 0.021] and pain catastrophizing [*p* = 0.003]), peripheral sensitization (lower pain thresholds), and worse motor function (balance test) (adjusted R^2^ = 36%, F (5, 99) = 12.57, *p* < 0.0001). Interestingly, no TMS or EEG markers were associated with VAS pain. **Conclusions:** Our study supports the notion that pain during physical activity is associated with a neural signature that demonstrates a lack of compensatory mechanisms for pain (decreased cortical inhibition, higher beta-band oscillations, and defective CPM), and it is different from the pain at rest, measured by the VAS, which is related mostly to emotional circuit dysregulation. These findings are important for developing better-targeted neural therapies given the contribution of different neural mechanisms to OA pain.

## 1. Introduction

One of the most common causes of chronic pain and disability in individuals over the age of 65 is knee osteoarthritis (KOA) [1]. Osteoarthritis in general affects over 7.6% of the global population, the prevalence is higher in women, and given the aging of the population worldwide, its prevalence is expected to increase significantly [2,3,4,5,6]. The main risk factors for KOA include female sex, lower socioeconomic status, obesity, sports participation and having a previous knee injury [4,7,8,9]. A common occurrence in individuals with KOA is a symptomatic mismatch when it comes to pain and disease severity: the amount of pain felt is not justified by the disease stage [10]. For this reason, and the lack of full understanding of the chronic pain mechanisms in OA, the conventional, pharmacological treatments for KOA pain are usually ineffective or unsatisfactory with several side effects [11,12]. Thus, it is important to fully comprehend the factors behind the dissonance of KOA symptomatology and its severity through the identification of predictors of KOA pain intensity.

As the most common manifestation of KOA, chronic pain is at the center of treatment and the main reason behind patients with OA seeking clinical attention [13]. The context of chronic pain in OA has been found to be caused by peripheral and central mechanisms and to be affected by multidimensional factors, proposing a biopsychosocial model of pain [14,15]. The peripheral mechanisms of KOA pain include the reduction in the pain threshold, and thus, the hypersensitivity of nociceptors found in the joint, mediated by inflammatory molecules such as prostaglandins and different cytokines [16,17]. Once the painful signals reach the central nervous system, the central mechanisms of chronic KOA pain are related to the process of central sensitization, which includes the lowering of central pain threshold recognition and the causing of pain to be more perceptible through even innocuous or unpainful stimuli, leading to a common phenomenon in patients with KOA, which is the disease severity and pain intensity dissociation [17]. Moreover, considering the central mechanisms of chronic pain in OA, including central sensitization and malfunctioning cortical excitability networks, evidence indicates that the intracortical inhibition and endogenous pain modulation system significantly impact the modulation and maintenance of chronic pain in KOA [18]. However, the concrete pain mechanisms and central nervous system influences on chronic KOA pain have yet to be fully comprehended. Given this understanding, these systems can lead to a more mechanism-driven treatment for this condition. For instance, more conservative approaches to treating OA pain, such as patient education, manual therapy, and dry needling for pain management, can target the extra-peripheral pain mechanisms in individuals with KOA pain, which could help reduce the dissociation between disease progression and pain intensity due to its multimodal approach [19,20,21].

Several studies have found associations between sleep patterns, fatigue, depression, and catastrophizing, and pain in individuals with OA [22,23,24]. Not only is pain increased by these factors, but individuals’ responses to treatment for KOA chronic pain are suggested to be modulated by these biopsychosocial factors. Rayahin and colleagues have found a significant reduction in the KOA pain treatment efficacy in patients with high pain catastrophizing thresholds as well as low self-efficacy perceptions [25]. To this point, other studies have linked lifestyle habits and found sociodemographic factors to be increasingly related to the manageability of KOA chronic pain [26,27]. These studies further suggest that current KOA pain treatments could be overlooking factors that are directly linked to the diminishing of pain and the disability this condition can cause to its patients when solely focusing on the peripheral mechanisms of this pain, thus suggesting that there is a need for a multimodal approach, specifically focusing on factors that are provenly associated with the chronicity of KOA pain.

Few studies have investigated the predictors of pain outcomes in chronic pain conditions, including OA [28,29,30]. Identification of psychosocial factors such as depression and pain catastrophizing as predictors of chronic pain has been reported and suggests a further understanding of pain chronicity [31]. Although there have been efforts to identify clinical predictors of chronic KOA pain, only a few studies used a multimodal assessment including not only clinical but also neurophysiological biomarkers as potential KOA pain predictors. Unlike psychosocial predictors of pain, the results concerning objective physiological metrics are contradictory. For example, one study [32] reported that intracortical inhibition is negatively correlated with pain intensity (assessed by the VAS) but not conditioned pain modulation (CPM) in KOA patients. On the other hand, Tavares and colleagues 49 investigated the explanatory predictors of pain intensity in older adults with KOA. They reported that a low CPM response, a low Von Frey threshold, higher anxiety, and worse radiological severity were predictors of pain intensity (assessed by the Brief Pain Inventory). However, there were no neurophysiological assessments in that study.

Therefore, given the need for a better understanding of the factors driving chronic pain in individuals with KOA, we propose a multivariate model investigation of the possible multimodal factors (psychosocial, quantitative sensory testing [QST], transcranial magnetic stimulation [TMS], and electroencephalography [EEG] variables) that predict the chronic pain intensity in patients with KOA. The objective of this paper is to use multivariate regression models to identify not only biopsychosocial but also neurophysiological predictors of pain intensity when associated with scales such as the WOMAC pain scale and the visual analog scale. Additionally, we test the differential predictors for activity-related pain (WOMAC pain) and non-specific pain (VAS) assessments. Given these two outcomes measure the pain associated with a different neural status (during functioning and resting), this will provide interesting insights in the neural mechanisms associated with pain.

## 2. Materials and Methods

### 2.1. Study Design

This study consists of a cross-sectional analysis of patients with knee OA who participated in a prospective cohort study titled “Deficit of Inhibition as a Marker of Neuroplasticity (DEFINE study) in rehabilitation” [33]. In this study, several biopsychosocial and neurophysiological markers were collected for patients with different conditions, such as KOA, while they underwent a rehabilitation program. The DEFINE protocol and this study were approved by the Research and Ethical Committee of Hospital das Clínicas da Faculdade de Medicina da Universidade de São Paulo (HC FMUSP) (registration number: 86832518.7.0000.0068). All the proceedings and methods in this study are in accordance with Brazilian research ethics regulations and the Declaration of Helsinki.

### 2.2. Participants

The DEFINE cohort included adults (over 18 years old) with a clinical and radiological diagnosis of knee OA (magnetic resonance imaging or computerized tomography, or bilateral knee radiography), clinical stability verified by medical evaluation, a written informed consent form signed by the subject, and who met the eligibility criteria for the Instituto de Medicina Física e Reabilitação (IMREA) rehabilitation program [31]. We excluded subjects if they were pregnant, had active OA clinical manifestations in joints other than the knee, or if they had any other clinical or social conditions that could interfere with the patient’s participation in the rehabilitation program [33].

In total, 100 individuals with clinically diagnosed KOA were included in the DEFINE protocol. The sample size was defined by previous studies with smaller cohorts of 35 and 55 patients that were powered to yield significant correlates with similar, but fewer, variables associated with the studied conditions in the protocol [33,34,35]. Furthermore, to reduce the bias in terms of the pain assessment, patients selected to be in this study had roughly the same number of years of chronic pain, had underwent the same treatment procedures and were not undergoing any clinical or novel treatments for their pain. For further participant descriptions, please refer to the DEFINE cohort study protocol [33].

### 2.3. Study Procedures

We invited patients admitted to the IMREA’s conventional rehabilitation program with knee OA to participate in this study. Interested patients signed the informed consent form. During one visit, a trained researcher performed a series of clinical and neurophysiological assessments in a standard format. We selected the instruments to enable a global assessment of KOA patients. Data for this study were collected from the DEFINE protocol from July 2020 through November 2021.

### 2.4. Demographic and Clinical Assessments

Investigators collected information regarding the participants’ age, gender, time of ongoing pain, height, weight, and body mass index from a standardized medical interview. In addition, to characterize this study’s sample, we performed a multidimensional assessment using standardized scales, including cognition (MOCA scale), emotional functions (Hamilton depression scale, hospital anxiety and depression scale), balance by Berg balance scale (BBS), and pain catastrophizing (pain catastrophizing scale). A summary of all the assessments can be seen in Appendix A.

### 2.5. Pain Intensity

We assessed the pain intensity using the visual analog scale (VAS, non-specific pain assessment) and the Western Ontario and McMaster Universities Osteoarthritis Index (WOMAC) pain scale (activity-related pain assessment). The VAS consists of a 10 cm straight line on a piece of paper. On one of its ends is the phrase “no pain” on zero centimeters and the other has “maximum pain” on ten centimeters. We asked patients to mark their discomfort level on the VAS line. The instructions for the patient were “Identify the amount of pain experienced in the last 48 h and make a mark perpendicular to the ‘no pain’—‘maximum pain’ line” [36]. In addition, the WOMAC pain subscale assesses pain according to 5 items: during walking, using stairs, in bed, sitting or lying, and standing upright. The investigator asked the subject to score the pain as none (0), mild (1), moderate (2), severe (3), or extreme (4) [37]. These scales were selected to assess the pain intensity and perceptions of the subjects participating in this study, given their broad encompassing of subjective pain perception within different actions and resting states. This is especially important to consider when assessing pain in KOA patients when we consider the peripheral and central mechanisms of this type of chronic pain as well as the pain’s intensity dissociation with disease progression.

### 2.6. Static and Dynamic Quantitative Sensory Testing (QST)

Quantitative sensory testing has been one of the more broadly accepted assessments of central sensitization within the field of chronic pain research. It is especially helpful in understanding the pain perception of patients who have maladaptive plasticity because of chronic pain conditions such as KOA. Given its context and wide utility in the chronic pain field, it would be significant to assess its importance as a predictor of KOA pain intensity and perception, which is the reason for its inclusion as one of the assessed predictors.

### 2.7. Pressure Pain Threshold (PPT)

We defined the minimum amount of pressure that triggers pain in pre-established regions (thenar region, and the region located one inch above the knee) using an algometer [38]. Also, we performed three algometry measurements (15 s intervals) and calculated the average.

### 2.8. Conditioned Pain Modulation (CPM)

We assessed the CPM response as a measurement of pain processing and, through intense heterotopic stimulation, of the response of the descending pain inhibitory system [39,40]. According to previous protocols [41,42], we used a PPT-dependent CPM protocol (the degree of pain modulation is assessed by changes in the PPTs). We asked the subjects to immerse one of their hands in a receptacle of cold water (10–12 °C) for one minute. After 30 s of immersion, the investigator presented the visual analog scale (VAS) to patients to indicate their pain level, referring to the submerged hand. Subsequently, we took three algometric measures (PPTs) (spaced between 15 s) for the contralateral hand. After an interval of approximately 10 min (time for the hand to return to normal body temperature), the subject immersed the other hand in the receptable and followed the previously stated protocol [42]. The team calculated the CPM response as the difference between the average PPTs minus the average PPTs during the conditioned stimulus. Additionally, for modeling purposes, we categorized the CPM response as a defective (<10% of changes) or effective (≥10% of changes) response, based on our previous results [14,19].

### 2.9. Transcranial Magnetic Stimulation (TMS)

Transcranial magnetic stimulation has been a widely studied neurophysiological assessment for the motor cortex and its implications for different chronic pain conditions. Considering that the central, maladaptive mechanisms of chronic KOA pain are deeply connected to the motor cortex, specifically the M1 region, this is an interesting neurophysiological assessment to be evaluated as a predictor of this condition’s pain intensity.

We used the Magstim Rapid^®^ stimulator (The Magstim Company Limited, Whitland, UK) to assess the TMS measurements. For that, we placed a 70 mm coil in a figure of eight at 45 degrees of the scalp to send a perpendicular pulse over the right and left motor cortex (for all the assessments). The assessor managed the coil’s stability and direction without neuronavigation. We recorded the muscular response to the stimulus using surface electromyography (EMG) with Ag/AgCl electrodes positioned on the first dorsal interosseous (FDI) muscle of the hand and the grounding electrode positioned on the wrist [43].

We performed a bilateral upper limb assessment and used anatomical references for motor cortex localization. Initially, we identified the vertex (intersection between the nasion–inion lines and zygomatic arches); then, we made a mark 5 cm from the vertex toward the ear tragus in the coronal plane. We determined the hotspot as the location with the highest and most stable motor evoked potential (MEP) amplitudes over the FDI. The resting motor threshold (rMT) was defined as the minimum intensity necessary for a single TMS pulse on the hotspot to generate an MEP, with at least a 50 μV peak to peak amplitude, in 50% of attempts [44]. We performed the following measures: MEP (intensity at 120% of the rMT, we calculated the peak-to-peak amplitude) and cortical silent period (CSP), which represents the temporary suppression of electromyographic activity during a sustained voluntary contraction. Moreover, we performed paired-pulse protocols of intracortical inhibition (SICI), assessed by interstimulus intervals of 2 ms, and intracortical facilitation (ICF) assessed by 10 ms interim stimulus intervals [44]. Ten randomized stimuli were applied at each interval and the average were calculated.

For the TMS neurophysiological measurements, we pooled the rMT, CSP, SICI, ICF, and MEP results from each hemisphere to obtain a bi-hemispheric average. This approach can be justified due to the bi-hemispheric nature of pain perception [45]; in addition, most of the sample were patients with bilateral knee OA. The TMS data were recorded and stored in a computer for offline analysis.

### 2.10. Resting-State Electroencephalography (EEG)

#### EEG Acquisition

We recorded the EEG following a standardized approach [46] in a quiet room. Assessors asked the patients to sit comfortably, have their sight directed naturally below the horizon line, not move or talk, and relax as much as possible. The investigator made sure they did not fall asleep by observing the patients and verbally calling their attention if drowsiness was noticed. We recorded the resting-state EEG for 5 min with the eyes closed using a 128-channel EGI system (Electrical Geodesics, Inc.) (EGI, Eugene, OR, USA). The EEG was recorded with a band-pass filter of 0.3–200 Hz and digitized at a sampling rate of 250 Hz. The EEG assessment was divided into a resting and task-related condition. The task-related assessment lasted 8 min, and the resting lasted a total of 10 min (5 min with eyes open and 5 min with eyes closed). The task-related condition included tasks such as movement observation, movement imagery, and movement execution. Band oscillations from the motor cortex during these conditions were recorded through MATLAB and interpreted by a specialist clinical neurophysiologist. For further description of the EEG data analysis, please refer to the DEFINE protocol [33].

### 2.11. Resting-State Spectral Power Analysis

We exported the data for offline analysis with EEGLab [47] and MATLAB (MATLAB R2012a, The MathWorks Inc., Natick, MA, USA, 2000). The EEG was re-referenced to the average; we used finite impulse response filters, one high-pass filter of 1 Hz and a low-pass filter of 50 Hz, followed by manual artifact detection and rejection by a blinded assessor to exclude the existence of any signal of drowsiness (attenuation of the alpha rhythm), epileptiform or any abnormal discharges prior to admission into the full study (no epileptiform or abnormal discharges were found). This analysis was followed by a manual artifact detection and rejection and independent component analysis (ICA); finally, we removed the ICs associated with artifacts and reconstructed the signal [48]. We processed the artifact-free data using the pop_spectopo EEGLab function with fast Fourier transformation with 5 s windows with a 50% overlap. We calculated the absolute power (μV2) and relative power (power in a specific frequency range/total power from 1 to 40 Hz) for the following frequency bands: delta (1–4 Hz), theta (4–8 Hz), alpha (8–13 Hz), and beta (13–30 Hz), and the following sub-bands: low beta (13–20 Hz) and high beta (20–30 Hz). We calculated all the EEG-related measurements from three regions of interest (ROIs): the central, parietal, and frontal areas, since they are important cortical regions involved in pain perception [49]. Also, we selected and averaged the electrodes representing these regions (the electrode placement is presented in Appendix A). Further information on the EEG data analysis can also be found in the DEFINE study’s published protocol [33].

### 2.12. Statistical Analysis

We used descriptive statistics to report the baseline characteristics. Continuous data were expressed as the mean and standard deviation (SD) or as the median and interquartile ranges, depending on their distribution. Dichotomous and categorical data were described in frequencies and the respective percentages. The histogram and Shapiro–Wilk test assessed the data distribution for normality. We labeled as outliers the values greater than 3 SDs away from the mean scores of the dependent or independent variables. After determining that the data had a sufficiently normal distribution, we conducted exploratory multivariate linear regression models to identify the relationships between the pain intensity according to the VAS and WOMAC pain scales (dependent variables) and the demographics, clinical, QST, resting EEG spectral power values, and TMS variables (independent variables). First, to select the best explanatory covariates, we created a correlation matrix and univariate linear models with each independent variable to detect significant covariates for an alpha level of 0.2. Variables that were not significant in the univariate models were eliminated. As a second step, a model was created with all the variables below the significance level (*p* < 0.2) in the univariate models. Thirdly, we checked the regression coefficients for significance and excluded those with a *p*-value > 0.05 from the model. Finally, to select our final multivariate modes, we search for confounders using a multicriteria approach: (i) based on the previous literature supporting physiological plausibility, (ii) considering changes in the β coefficients of more than 10%, and (iii) using the Akaike information criteria to select the variables that would result in the best fit [50,51]. A backward stepwise model was utilized for the variable selections to ensure the most optimal model was created with the highest R^2^ possible to explain the pain prediction models [52]. We also tested the interaction of the demographic and clinical variables with the main predictors’ variables, which was included in the final models if significant. Age, sex, and disease severity (indexed by the Kellgren–Lawrence classification) were explored as biological variables that could potentially confound all the final models. Once the final model was determined, we added these variables as covariates, and if not significant, they were excluded from the models.

We assessed the assumption of linearity by visually comparing the scatterplot of each independent variable and a superimposed regression line. The assumption of homoscedasticity was checked by visual inspection of the scatterplot of the standardized predicted values and standardized residuals [53]. The residuals were tested for normality using histograms and the Shapiro–Wilk normality test [54]. The Durbin Watson estimates and Cook’s distances were used for analysis of regression diagnostics such as multicollinearity and influential cases.

We used R version 4.0.2. for the statistical analyses [55]. Because this was an exploratory study and to minimize the risk of type II errors, no correction for multiple comparisons was performed.

## 3. Results

### 3.1. Sample Characteristics

We included 105 patients with chronic knee OA. Bilateral symptoms were reported by most participants (99.02%). The majority of the participants were female, older adults, and overweight or obese. The average pain was moderate (WOMAC pain score of 10.77 [SD = 4.18] from 0 to 20 on the scale; and VAS pain of 5.53 [SD = 2.06] from 0 to 10 on the scale). A detailed description of the clinical and neurophysiological characteristics is provided in Table 1.

### 3.2. Univariate Analysis

We found a statistically significant moderate correlation between the activity-related (WOMAC pain) and non-specific (VAS) pain assessment (correlation coefficient = 0.63, R squared = 0.39, *p* < 0.0001). Both pain assessments were negatively associated with a higher education level, cognition score (MOCA test), walking distance for the six-minute walking test (for the activity-related assessment and physical pain comparability), quality of life score (SF-36), and upper limb pain threshold (applicable for the identification of central pain nociception comprehension). Similarly, both pain assessments were positively associated with the pain catastrophizing score, anxiety level (HAD scale), depression (Hamilton scale), sleepiness score (Epworth scale), disability score (WOMAC total score), and radiographic disease severity (Kellgren–Lawrence classification). Moreover, we found some associations that were statistically significant in relation to only one of the pain assessments. The WOMAC pain was negatively correlated with the knee pain threshold and the theta-band relative power in the frontal, central, and parietal areas. Conversely, the VAS pain was positively associated with the time during the 10 m walk test and negatively associated with the balance test score (Berg scale). A detailed description of the univariate models is available in Appendix A.

### 3.3. Multivariate Analysis

#### 3.3.1. Predictors of WOMAC Pain Score

##### General Findings

We found that the WOMAC pain score is positively correlated with the depression scale (β = 0.275, 95% 0.174 to 0.376; *p* < 0.001), pain catastrophizing (β = 0.126, 95% CI: 0.074 to 0.178; *p* < 0.001), and SICI (β = 1.067, 95% CI: 0.887 to 3.022; *p* = 0.021; Figure 1A). Furthermore, the upper limb pain threshold (β = −0.487, 95% CI: −0.759 to −0.215; *p* = 0.001; Figure 1C) and effective CPM response category (β = −2.200, 95% CI: −3.289 to −1.110; *p* < 0.001; Figure 1D) are portrayed as having a statistically significant, negative association with the WOMAC pain (Table 2, Model 1A). The multivariate model 1A resulted in the following final regression equation: WOMAC pain score = 10.014 + (0.275 × depression) + (0.126 × pain catastrophizing) + (1.067 × SICI) + (−0.487 × upper limb pain threshold) + (−2.200 × effective CPM response category).

### 3.4. Findings Summary

Based on the findings from the multivariate models, we hypothesized the differential importance of pain-related domains in the WOMAC and VAS pain ratings (Figure 2). The WOMAC score is driven mainly by the sensory–motor component (indexed by the SICI, beta-band oscillations, and CPM), indicating a possible reason why neurophysiological findings such as the EEG predictors are significant specifically for the WOMAC pain scale perception. On the other hand, the VAS score is associated similarly with all the components, becoming non-specific and highly variable and furthermore related to individual pain perception and psychosocial factors.

### 3.5. Neurophysiological Association Findings

Interestingly, from the patients with EEG data available (n = 66), we found collinearity between the low-beta-band relative power in the frontal areas and the SICI; the correlation coefficient was 0.85 (moderate correlation, *p* = 0.03), and the variance inflation factor was 3.60. Therefore, we fitted a separated multivariate model to assess the independent association of the EEG spectral power with the WOMAC pain. We found a positive correlation between the low-beta-band relative power in the frontal areas (β = 14.149, 95% CI: 1.679 to 26.619; *p* = 0.027: Figure 1B) and the WOMAC pain intensity (Table 2, Model 1B).

### 3.6. Model Variance Explanation

Patients with higher WOMAC pain intensity had higher depression and catastrophizing scores, lower intracortical inhibition, a lower upper limb pain threshold, a defective CPM response (<10% changes from baseline), and higher beta-band power in the frontal areas. The multivariate Model 1B (n = 66) resulted in the following final regression equation: WOMAC pain score = 7.888 + (14.149 × low-beta-band relative power) + (0.100 × pain catastrophizing) + (0.296 × depression) + (−0.532 × knee pain threshold) + (−2.159 × effective CPM response category).

The multivariate models statistically significantly predicted the WOMAC pain scores (Model 1A: F (5, 99) = 22.81, *p* < 0.0001; 1B: F (5, 61) = 13.83, *p* < 0.0001) and included successful multimodal assessments. The models were able to explain 52 to 57% of the variance in the WOMAC pain scores.

### 3.7. Predictors of VAS Pain Scale

#### 3.7.1. General Findings

In relation to the VAS scale, similarly to the WOMAC pain, depression (β = 0.093, 95% CI: 0.014 to 0.172; *p* = 0.021), and pain catastrophizing (β = 0.050, 95% CI: 0.018 to 0.083; *p* = 0.003) were positively correlated with the pain intensity, while the upper limb pain threshold was negatively correlated with it (β = −0.145, 95% CI: −0.304 to −0.014; *p* = 0.044). Furthermore, we found that the balance function (indexed by berg balance test) (β = −0.112, 95% CI: −0.156 to −0.068; *p* < 0.001) and anxiety (β = 0.077, 95% CI: 0.030 to 0.185; *p* = 0.037) were negatively associated with the VAS pain, but not with the WOMAC pain. Hence, patients with higher VAS pain had higher depression, anxiety, and pain catastrophizing, lower pain thresholds, and worse balance performance.

#### 3.7.2. Neurophysiological Findings

Unlike the WOMAC models, the SICI, CPM and EEG variables were not statistically significant associated with the VAS pain intensity. Our findings suggest that assessment of central pain inhibition variables can be more related to the movement perception and activity-related neural mechanisms of chronic pain, considering that a more generic, individually perceptive scale such as the VAS did not provide significant correlations with these pain inhibition markers.

#### 3.7.3. Model Variance Explanation

The final regression equation was as follows: VAS pain score = 10.656 + (0.093 × depression) + (0.050 × pain catastrophizing) + (−0.077 × anxiety) + (−0.145 × upper limb pain threshold) + (−0.112 × balance test). The multivariate models statistically significantly predicted the VAS pain scores (F (5, 99) = 12.57, *p* < 0.0001) and explained 36% of the VAS pain variability in our sample.

In both modeling processes for the WOMAC and VAS assessments, age, sex, time of ongoing pain, and OA severity (indexed by the Kellgren–Lawrence classification) were not confounders or effect modifiers; thus, they were not included in the final models. Also, the interactions between the clinical and physiological variables were not significant in the final models.

## 4. Discussion

### 4.1. Main Findings

In the present study, we aimed to explore the predictors of pain intensity as measured by the VAS and WOMAC pain scales, and the clinical and neurophysiological variables in patients with chronic pain due to KOA. Our main findings revealed important relationships between clinical and neurophysiological variables and pain intensity. From the multivariate models, we found different sets of predictors for the activity-related (WOMAC pain) and non-specific (VAS) pain assessments. (1) A higher WOMAC pain score is predicted by sensory–motor markers (lower intracortical inhibition and higher beta-band oscillations) and central sensitization (dysfunctional CPM response), in addition to the psychological and peripheral sensitization factors. (2) Conversely, higher VAS pain intensity is only predicted by psychological factors (higher depression, anxiety, and pain catastrophizing), peripheral sensitization (lower pain thresholds), and worse motor function (balance test). Interestingly, no TMS or EEG markers are associated with the VAS pain in our univariate or multivariate models. Noteworthily, our multimodal approach yields a highly predictive model (adjusted R-squared around 52 to 57%), showing the potential of combining multidomain assessments to understand pain perception. Moreover, these predictors used in understanding pain assessment scores can lead to more individualized and thus more effective treatment models for those with chronic KOA pain. To our knowledge, this is the first study to combine clinical, radiological (X-ray), sensory (QST), and neuropsychological data (TMS and resting-state EEG) to predict pain in chronic KOA patients.

### 4.2. Dissociation of Neural Mechanisms Associated with WOMAC and VAS Pain Scores

One important result is the different set of predictors when comparing the WOMAC pain and VAS pain scores. In fact, these two variables are moderately correlated in our sample, and indeed, the WOMAC pain explains less than 40% of the variability of the VAS pain. This highlights the multidimensional nature of pain perception and the importance of choosing the most appropriate pain outcomes in clinical practice and research [56]. We found that higher activity-related pain (indexed by WOMAC pain) is predicted by lower intracortical inhibition, higher beta-band oscillations, and defective CPM. However, the VAS pain was not associated with that neural signature. These findings align with our previous studies, where we did not find a correlation among the VAS pain and ICI/CPM [34,57,58,59]. Additionally, a recent meta-analysis did not find any correlation between CPM and clinical manifestations of chronic pain. Noteworthily, most of the studies analyzed used the VAS or “VAS like” scales to measure the pain intensity in their studies [60].

Therefore, we hypothesize that the multifactorial influence and the high variability of pain reduces the detection accuracy of the VAS pain. This is supported by previous studies showing that the VAS is prone to context bias and has high disagreements with comprehensive scales [61]. These limitations may explain the lack of relationship between more direct sensorimotor biomarkers, such as the ICI, beta oscillations, and CPM, and the VAS pain (Figure 2). On the other hand, the WOMAC pain scale assesses the pain perception within a fixed context—daily life activities (walking, resting, standing, sleeping, climbing stairs)—hence, it seems to be more stable, less prone to context bias, and directly associated with sensory–motor function and its related biomarkers (Figure 2). The neural signature we found to be associated with higher activity-related pain (WOMAC) shows a likely disrupted compensatory neural mechanism and thus explains the higher pain during daily functioning as measured by the WOMAC scale.

Due to the complexity of validating self-reported pain scales, it is not possible to develop a pain outcomes hierarchy. Thus, we suggest not using only one pain measurement in clinical practice or research, and if possible, performing a sensitivity analysis by pain outcome, especially when analyzing clinical trials and predictive models.

### 4.3. Cortical Inhibitory Markers: Intracortical Inhibition and Frontal Beta Oscillations

We found a positive association between the frontal beta-band oscillations and the WOMAC pain intensity. Since we observed negative collinearity between the frontal beta oscillations and the SICI, meaning that higher low-beta-band oscillations were statistically similar to low intracortical inhibition in the models, we hypothesize that higher beta-band oscillations may represent a maladaptive compensatory excitatory mechanism produced in the frontocentral cortex. Therefore, the presence of higher beta-band power in this region would be equivalent to a state of intracortical disinhibition detected in the primary motor cortex, which studies have found to be related to pain intensity [62]. However, we did not find the same association with the VAS. In this context, the increased beta seems to be related to a compensatory mechanism of greater neuronal injury, which has been reported in KOA patients to be expressed as neuropathic symptoms [63]. This finding can also be seen in other neural injury examples, such as in spinal cord injury [34,64] and stroke [35,65]. In fact, frontocentral beta oscillations seem to be related to increased local brain activity [66,67], thus being likely in KOA patients to generate additional oscillatory activity to compensate for the OA indirect neural lesion, joint inflammation, and peripheral sensitization. When looking at studies on musculoskeletal pain conditions, our results agree with previous reports on chronic hip OA [68] and lower back pain [69], where higher frequency brain oscillations in the frontal areas are associated with higher pain intensity. Comparatively, whereas these studies have found this association between numerical rating scales (NRSs), they did not account for other biopsychosocial factors that may have interfered with our finding of neurophysiological predictors with such a non-specific pain scale such as the NRS [68,69]. Moreover, our models proved highly explicative of the pain perception between our two scales, because they encompass so many significant predictors, whereas other studies that have found similar results have lower explained percentages with their predictors.

Moreover, we found that the activity-related pain intensity is associated with less intracortical inhibition (higher SICI). It has been reported that chronic pain presents maladaptive neuroplasticity in the motor cortex, evidenced by the altered corticospinal and lower intracortical inhibition when compared to healthy participants [62]. However, the association between the SICI and the pain intensity is not yet completely understood. Similar to our study, Tarragó et al. [18] reported a negative correlation between the cortical inhibition and the pain intensity (assessed by the WOMAC scale) in chronic KOA patients. Since the SICI assesses the cortical GABAergic transmission (GABA-A) [70], we hypothesize that sustained pain—in this case, prompted by knee inflammation—could lead to a sequence of compensatory events (including the use of drugs that could disrupt effective compensatory mechanisms), resulting in a dysfunctional inhibitory function. Thus, unbalanced GABAergic and glutamatergic intracortical networks might explain the relationship between the anomalous disinhibition and the higher activity-related pain intensity in chronic KOA patients. Future studies should address the clinical effects of modulatory interventions to restore this inhibitory dysfunction.

### 4.4. Quantitative Sensory Testing (QST)

In our study, we found that deficient conditioned pain modulation is associated with the WOMAC pain intensity. This relationship corroborates previous findings that the CPM is inefficient in chronic pain conditions [71,72,73,74], being related to the pain intensity [72,75] and central sensitization [76]. Therefore, our results support previous findings that the CPM plays an important role in measuring the endogenous pain modulation system. Accordingly, this system is a key component of the chronicity of the pain, being an important target in the development of interventions for the chronic pain population. Furthermore, we found in all our models a negative relationship between the PPT and the pain intensity. These findings are consistent with the idea that the pain threshold can provide information about peripheral and central sensitization [77,78]. Previous research has shown associations between pain intensity and pain thresholds by algometry in chronic pain populations [79,80,81], and it being a predictor of the treatment response [82]. Taken together, the available evidence opens up new treatment approaches, with the possibility for precision pain medicine treatment according to pain phenotyping. With this view, future studies should explore the possibility of matching non-pharmacological and pharmacological strategies to the central sensitization phenotypes indexed by the altered CPM and PPT.

### 4.5. Depression

All our models revealed a positive correlation between depression and pain intensity in patients with chronic KOA pain. Previous studies corroborated these findings, showing depression to be associated with pain intensity in several chronic pain syndromes [83,84,85,86], such as knee and hip OA [87], lower back pain [88], post-trauma [84], migraine [89], and neuropathic pain populations [90]. The relationship between pain and depressive behavior is already well established in the literature, although the psychological profile may be secondary to the pain persistence over time or the cause itself [91,92]. Therefore, mood disorders seem to be related to the whole pain chronicity process, from being a risk factor to a consequence of chronic pain. Adequate depression screening and mental health counseling should be offered as early as possible to chronic KOA pain patients and must also be part of integrative KOA management.

### 4.6. Pain Catastrophizing

Moreover, we detected in our models a positive relationship between catastrophizing and pain intensity according to the VAS and WOMAC. These findings are aligned with previous studies that demonstrated an association between catastrophizing, pain intensity [87,93,94,95], and opioid prescription [94]. Outstandingly, we detected catastrophic thinking to be an independent factor for pain intensity, when controlled for depression. This result suggests that catastrophizing behavior may be related to pain even in non-depressed patients, as suggest before [96]. Consequently, pain-related catastrophic thinking and depression, although correlated, may measure different components of the psychological state of chronic pain patients and could be developed independently of depression or anxiety [30]. Therefore, pain catastrophizing should be prevented and addressed during KOA pain management as an important independent factor in terms of pain intensity.

### 4.7. Negative Findings

Interestingly, a few variables that were expected to have predictive associations with pain perceptions, such as gender, time of pain, and disease severity, were not found to be significant predictors in our model. In other studies assessing the gender differences in pain perception, women have been found to be more sensitive to chronic pain associated with knee osteoarthritis when compared to men [97]. However, these studies may not have accounted for other confounding variables that could have led to this difference in perception, such as pain catastrophizing and depression. Moreover, our study’s demographic is made up in its majority by women, thus possibly rendering gender an unsignificant predictor within our cohort.

The lack of association between the time of pain and disease severity and the pain perception in our study can possibly be explained by the central mechanisms of KOA pain. Our cohort of subjects consists of patients with several years of KOA pain; thus, it is homogeneous enough that their perception of pain may not be associated with the length of time they have been feeling pain. Moreover, a common phenomenon found in patients with chronic KOA pain is the dissociation between disease severity and pain intensity, mainly caused by the maladaptive mechanisms of central sensitization [98]. Central sensitization produces lower pain thresholds for different joint stimuli, painful or not, even if the disease has not thoroughly or aggressively progressed [17]. Therefore, considering our cohort has a significant number of patients with high central sensitization, it would make sense for the disease severity to not be such an intricate predictor of pain intensity within it.

### 4.8. Limitations

Although our findings provide significant and interesting results within the context of KOA pain’s predicting factors, our study is not devoid of limitations. Future studies including other measurements of OA severity, such as magnetic resonance imaging and vibroartrography, might be useful to analyze the differences in pain mechanisms according to the OA severity. For one, the cross-sectional aspect of our study prevents us from conclusively defining the variables associated with pain perception on different scales as predictors, despite their strong correlations. Future longitudinal studies might be carried out to further support our findings and set these variables as definitive predictors. Moreover, our cohort denotes a specific type of patient with chronic KOA pain, which could denote a potential selection bias of the DEFINE protocol rehabilitation program; therefore, our predictor-related findings should not be generalized through all the classifications of KOA pain patients. Nonetheless, we are confident that our 100-patient cohort represents a significant portion of KOA patients, thus maintaining the validity of the prediction models.

## 5. Conclusions

We performed multivariate models including multimodal data (clinical, sensory, and neurophysiological variables). The inclusion criteria were broad, with no restriction on the pain intensity or pain-related comorbidities, thus ensuring our models’ adequate external validity. Finally, we used low-cost assessments as predictors (clinical scales, QST, TMS, and EEG); therefore, the potential clinical translation of our models’ results is high.

Our study supports the idea that peripheral and central sensitization, dysfunctional sensory–motor inhibitory tonus, and psychosocial factors influence the pain perception intensity in chronic KOA patients. However, the factors contributing to pain are associated with whether the pain is assessed at resting or related to physical activity. A non-specific pain scale such as the VAS is significantly associated with psychological factors (depression, anxiety, and pain catastrophizing), peripheral sensitization (PPTs), and motor function (balance test). Conversely, activity-related pain (WOMAC pain) is predominantly associated with a maladaptive compensatory neural response characterized by sensory–motor (lower intracortical inhibition and higher beta-band oscillations) and central sensitization (dysfunctional CPM response) variables, in addition to psychological and peripheral sensitization factors. This result could help to design specific treatments targeting neural structures associated with different pain states. Furthermore, the identified predictors of pain intensity could be used for risk stratification or phenotyping of chronic KOA patients, thus opening up the possibility of precision pain medicine treatment. Moreover, our multivariate and multimodal analyses explained almost two-thirds of the pain variability scores, thus illustrating the vast potential of integrated multidomain assessments to predict chronic pain. Therefore, our findings can provide a significant explanation of the pain perception and occurrence in chronic KOA patients. These variables can help further individualize KOA pain treatments to ensure better management and efficacy, given that they can identify if higher pain scores are based on motor-related or psychosocial aspects of central pain mechanisms. Finally, our findings also support the urgent need for a multidimensional approach to KOA pain treatment, involving treatments addressing multiple central and peripheral targets.

## Figures and Tables

**Figure 1 jcm-14-03633-f001:**
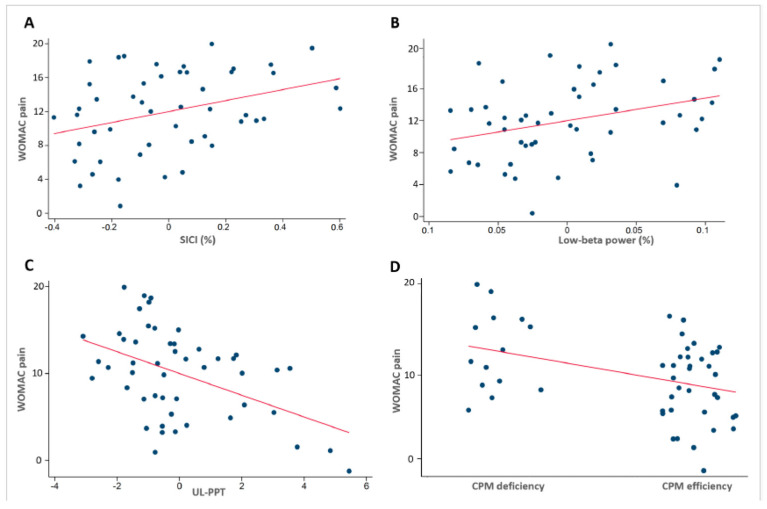
Adjusted correlations of the physiological markers and WOMAC pain from the multivariate models: (**A**) short intracortical inhibition (SICI, higher ICI represents lower intracortical inhibition), (**B**) low-beta-band relative power, (**C**) upper limb pressure pain threshold (UL-PPT), and (**D**) CPM response category. Regression lines were adjusted by the pain threshold, depression, catastrophizing score, and CPM response category.

**Figure 2 jcm-14-03633-f002:**
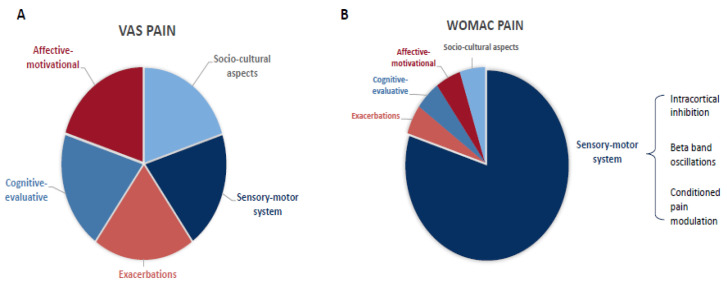
Theoretical representation of the influential components of the non-specific (VAS, (**A**)) and activity-related (WOMAC pain, (**B**)) pain intensity assessments.

**Table 1 jcm-14-03633-t001:** Baseline clinical and sociodemographic characteristics of the chronic knee OA participants.

Variables	Knee OA Subjects (N = 105)
**Demographics**	
Age	68.65 (9.45)
Gender (%)	
Female	90 (85.71)
Male	15 (14.29)
Ethnicity	
White	70 (66.66)
Black	11 (10.47)
Mixed	20 (19.04)
Asian	4 (3.83)
Education level (%)	
Illiterate	1 (0.95)
Elementary	45 (42.85)
High school	32 (30.47)
Superior	27 (25.73)
Weight (kilograms)	79.96 ± 15.56
Height (meters)	1.58 ± 0.09
BMI	31.99 ± 5.3
**Clinical assessments**	
Bilateral knee OA (%)	104 (99.02)
Time of ongoing pain (months)	95.74 ± 98.75
Total knee arthroplasty (%)	
Right	3 (2.86)
Left	3 (2.86)
Pain—visual analog scale	
Right	5.69 ± 2.83
Left	5.38 ± 2.79
Average	5.54 ± 2.06
WOMAC total score	50.84 ± 19.49
WOMAC pain	10.77 ± 4.18
WOMAC stiffness	4.55 ± 2.07
WOMAC physical function	35.52 ± 14.51
Kellgren–Lawrence classification	
Right	2.5 ± 1.19
Left	2.33 ± 1.18
Average	2.43 ± 1.15
Pain catastrophizing scale	14.26 ± 11.04
HAM-L scale	9.36 ± 5.58
Hospital anxiety and depression scale	
Anxiety	5.92 ± 4.26
Depression	4.23 ± 3.55
Montreal cognitive assessment	21.02 ± 5.03
Epworth sleepiness scale	10.20 ± 5.96
Quality of life (sf-36)—total score	53.69 ± 20.00
**Quantitative sensory testing**	
Pain pressure threshold	
Upper limb	
Right	5.84 ± 2.05
Left	5.56 ± 2.12
Average	5.69 ± 2.02
Knee	
Right	4.84 ± 2.57
Left	4.79 ± 2.49
Average	4.82 ± 2.49
Conditioned pain modulation	
Right	0.93 ± 1.42
Left	1.05 ± 1.45
Average	1.01 ± 1.29
Average (% of change from baseline)	19.51 ± 25.11
**Transcranial magnetic stimulation**	
Motor threshold	
Right	52.73 ± 11.59
Left	50.97 ± 11.00
Average	51.36 ± 11.45
Motor-evoked potential	
Right	1.76 ± 1.30
Left	1.87 ± 2.02
Average	1.81 ± 1.41
Cortico-silent period	
Right	91.84 ± 35.82
Left	80.81 ± 31.67
Average	86.33 ± 31.46
Short intracortical inhibition	
Right	0.46 ± 0.32
Left	0.49 ± 0.32
Average	0.47 ± 0.27
Intracortical facilitation	
Right	1.59 ± 0.65
Left	1.70 ± 0.82
Average	1.65 ± 0.58
**Electroencephalography (relative spectral power) (n = 66)**	
Frontal	
Delta	24.6% (IQR 17.1)
Theta	18.8% (IQR 10.5)
Alpha	25.2% (IQR 17.9)
Low beta	12.7% (IQR 7.4)
Beta	20.8% (IQR 13.7)
High beta	7.1% (IQR 7.2)
Central	
Delta	20.1% (IQR 15.5)
Theta	18.3% (IQR 9.8)
Alpha	26.6% (IQR 17.4)
Low beta	14.9% (IQR 8.7)
Beta	22.8% (IQR 14.7)
High beta	7% (IQR 7.8)
Parietal	
Delta	30.3% (IQR 24.0)
Theta	28.2% (IQR 33.9)
Alpha	57.8% (IQR 17.0)
Low beta	22.3% (IQR 27.8)
Beta	31.6% (IQR 35.7)
High beta	8.0% (IQR 11.8)

**Table 2 jcm-14-03633-t002:** Multivariate models of pain intensity.

Variables	Beta Coefficient	95% CI	*p*-Value	Adjusted R^2^
*Model 1A: WOMAC pain score*				0.519
Pain threshold—upper limb	−0.487	−0.759 to −0.215	0.001	
Hamilton depression scale	0.275	0.174 to 0.376	<0.001	
Pain catastrophizing scale	0.126	0.074 to 0.178	<0.001	
Patients with effective CPM response	−2.200	−3.289 to −1.110	<0.001	
SICI	1.067	0.887 to 3.022	0.021	
*Model 1B: WOMAC pain score*				0.572
Frontal low-beta relative power	14.149	1.679 to 26.619	0.027	
Pain threshold—knee	−0.532	−0.809 to −0.255	<0.001	
Hamilton depression scale	0.296	0.158 to 0.433	<0.001	
Patients catastrophizing scale	0.100	0.026 to 0.174	0.009	
Patients with effective CPM response	−1.456	−3.023 to 0.111	0.068	
*Model 2: VAS pain*				0.357
Hamilton depression scale	0.093	0.014 to 0.172	0.021	
Berg balance scale	−0.112	−0.156 to −0.068	<0.001	
Pain catastrophizing scale	0.050	0.018 to 0.083	0.003	
Pain threshold superior limb	−0.145	−0.304 to −0.014	0.044	
HAD anxiety scale	−0.078	−0.186 to −0.030	0.037	

## Data Availability

The original contributions presented in this study are included in the article/Appendix A. Further inquiries can be directed to the corresponding author.

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
