# Peer review of "Maladaptive Compensatory Neural Mechanisms Associated with Activity-Related Osteoarthritis Pain: Dissociation of Psychological and Activity-Related Neural Mechanisms of WOMAC Pain and VAS Pain"

_jcm, 2025, doi:10.3390/jcm14113633_

Round 1

Reviewer 1 Report

Comments and Suggestions for Authors

Dear Authors,

Thank you for the opportunity to review your manuscript, which addresses an important and relevant topic in the study of knee osteoarthritis (KOA) pain. Below, I provide detailed observations that could strengthen your research.

1. Title and Abstract

  • The title accurately reflects the study’s content, but it could be refined to better highlight the distinction between the neural mechanisms associated with pain assessed by WOMAC and VAS.
  • The abstract is well-structured, clearly stating the objectives and key findings. However, the statistical results could be presented more clearly to enhance readability. More precise differentiation between predictors of resting and activity-related pain is recommended.

2. Introduction

  • The background is well-founded, with an appropriate review of the literature on KOA, its impact on quality of life, and the underlying pain mechanisms.
  • The transition from the study rationale to the hypothesis could be improved. While the need for a multimodal approach is mentioned, its connection to previous studies could be reinforced with more recent references.
  •  In the Introduction section, when the authors mention conservative treatment, there are some work which can be very useful in mild-moderate OA. I recommend using and mentioning the following quality papers of studies investigating the use of Dry Needling and a 3-month program of Therapeutic Exercise, Manual Therapy, and Pain Education: doi:10.1093/pm/pnz036, doi:10.3390/app11041895, DOI: 10.3390/ijerph19106194

3. Methods

  • Study Design: The study is presented as a cross-sectional analysis within a larger cohort study. However, the participant selection criteria are not clearly described. It would be beneficial to specify the sample size justification based on a power analysis.
  • Variables and Measurements:
    • The use of multiple scales and biomarkers is a strength. However, the description of some measurement methods (e.g., EEG and TMS) could be more detailed to ensure reproducibility.
    • Clarification is needed on procedures used to minimize bias in pain assessment. Were repeated measurements performed? Were external factors such as medication use or comorbidities controlled?
  • Statistical Analysis:
    • A multivariate regression model was appropriately implemented to predict pain intensity. However, a clearer justification for variable selection and potential collinearity concerns should be provided.
    • It is not mentioned whether cross-validation of the model or sensitivity analysis was conducted, which could strengthen the robustness of the findings.

4. Results

  • The presentation of results is generally appropriate, though some descriptive data (e.g., demographic and clinical characteristics) could be better contextualized in clinical terms.
  • The comparison between WOMAC and VAS is central to the study, but the interpretation of regression coefficients should be further emphasized. In particular, the relationship between intracortical inhibition and perceived pain deserves a more detailed discussion regarding its clinical relevance.

5. Discussion

  • The findings are interpreted coherently, highlighting the differentiation between pain mechanisms at rest and during activity.
  • The clinical implications of these results should be emphasized more. How might these findings influence future therapeutic strategies?
  • The limitation of the cross-sectional design is acknowledged, but potential selection or measurement biases are not discussed.
  • There is an interesting work that develops to identify the differences in blood investigations between total hip and total knee replacement. I recommend the authors to discuss it: 10.1097/TGR.0000000000000337

6. Conclusions

  • The conclusions are sound but could better emphasize practical applications and future research directions.

Final Recommendations

  • Strengthen the methodological description, particularly regarding selection criteria and variable control.
  • Improve clarity in presenting statistical results and their clinical interpretation.
  • Expand on the therapeutic implications of the findings.
  • Consider additional analyses to reinforce the validity of predictive models.

Overall, the manuscript presents a well-designed study with relevant findings, but certain improvements in methodology and discussion could enhance its impact in the scientific literature.

Best regards,

Author Response

Reviewer 1 comments

Thank you for the opportunity to review your manuscript, which addresses an important and relevant topic in the study of knee osteoarthritis (KOA) pain. Below, I provide detailed observations that could strengthen your research.

  1. Title and Abstract
  • The title accurately reflects the study’s content, but it could be refined to better highlight the distinction between the neural mechanisms associated with pain assessed by WOMAC and VAS.
    • Authors: Thank you for your comment, we have added the significant neural mechanisms associated with each pain scale to the title for better distinctions. Changes and additions are in red font.
  • The abstract is well-structured, clearly stating the objectives and key findings. However, the statistical results could be presented more clearly to enhance readability. More precise differentiation between predictors of resting and activity-related pain is recommended.
    • Authors: Thank you for your comment, we have distinguished resting and activity-related pain predictors in the abstract as well as added the p-values for the significant associations to enhance readability. Changes and additions are in red font.
  1. Introduction
  • The background is well-founded, with an appropriate review of the literature on KOA, its impact on quality of life, and the underlying pain mechanisms.
  • The transition from the study rationale to the hypothesis could be improved. While the need for a multimodal approach is mentioned, its connection to previous studies could be reinforced with more recent references.
    • Authors: Thank you for your comment. We have added current articles that convey the need for a multimodal approach to chronic OA pain to strengthen the transition to our hypothesis. Changes and additions are in red font.
  •  In the Introduction section, when the authors mention conservative treatment, there are some work which can be very useful in mild-moderate OA. I recommend using and mentioning the following quality papers of studies investigating the use of Dry Needling and a 3-month program of Therapeutic Exercise, Manual Therapy, and Pain Education: doi:10.1093/pm/pnz036, doi:10.3390/app11041895, DOI: 10.3390/ijerph19106194
    • Authors: Thank you for your comment. The article recommendations are deeply insightful, and although the aim of this study isn’t focused on treatment of KOA, but the associated factors that could be managed by it, we have briefly added a connection on how these conservative treatments can be a gateway on focusing on these factors as a multimodal approach. Changes and additions are in red font.
  1. Methods
  • Study Design: The study is presented as a cross-sectional analysis within a larger cohort study. However, the participant selection criteria are not clearly described. It would be beneficial to specify the sample size justification based on a power analysis.
    • Authors: Thank you for your comment. As this analysis is derived from results of our larger cohort study, the DEFINE cohort, our sample size is derived from their power analysis, in which they have overestimated the need of patients to find the desired correlations, based on previous studies that were well powered to find similar results with less patients (35 and 55 patient cohorts). We have added this information to the participants section of our methods to better clarify this study’s sample size. Changes and additions are in red font.
  • Variables and Measurements:
    • The use of multiple scales and biomarkers is a strength. However, the description of some measurement methods (e.g., EEG and TMS) could be more detailed to ensure reproducibility.
      • Authors: Thank you for your comment. We have added more details on the TMS and EEG procedures to ensure reproducibility and have further referred readers to the DEFINE protocol published in Frontiers, in which these protocols are thoroughly described. Changes and additions are in red font.
    • Clarification is needed on procedures used to minimize bias in pain assessment. Were repeated measurements performed? Were external factors such as medication use or comorbidities controlled?
      • Authors: Thank you for your comment. We have clarified the procedures used to minimize selection bias and pain assessment bias in the participants section of the Methods. Changes and additions are in red font.
  • Statistical Analysis:
    • A multivariate regression model was appropriately implemented to predict pain intensity. However, a clearer justification for variable selection and potential collinearity concerns should be provided.
      • Authors: Thank you for your comment. Our regression model followed a backward stepwise approach. We have referred readers to the article that thoroughly describes how this approach avoids potential collinearity and enhances variable selection from our DEFINE cohort. Changes and additions are in red font.
    • It is not mentioned whether cross-validation of the model or sensitivity analysis was conducted, which could strengthen the robustness of the findings.
      • Authors: Thank you for your comment. We have noted how the cross-validation of the model has been done to further enhance credibility, given this model has yielded, dependable results in previous, interconnected analyses to the one in our current paper. Changes and additions are in red font.
  1. Results
  • The presentation of results is generally appropriate, though some descriptive data (e.g., demographic and clinical characteristics) could be better contextualized in clinical terms.
    • Authors: Thank you for your comment. We have further added descriptors that correlate the assessed variables with their potential clinical applicability. Changes and additions are in red font.
  • The comparison between WOMAC and VAS is central to the study, but the interpretation of regression coefficients should be further emphasized. In particular, the relationship between intracortical inhibition and perceived pain deserves a more detailed discussion regarding its clinical relevance.
    • Authors: Thank you for your comment. Under the “Neurophysiological results” section of our results we have added a couple sections explaining the clinical relevance of the relationships between different activity-related and psychosocial variables to each pain measurement scale. Changes and additions are in red font.
  1. Discussion
  • The findings are interpreted coherently, highlighting the differentiation between pain mechanisms at rest and during activity.
  • The clinical implications of these results should be emphasized more. How might these findings influence future therapeutic strategies?
    • Authors: Thank you for your comment. We have added a couple sentences at the first paragraph of our discussion as well as in the conclusion of our study, further explaining how these associations can purposefully connect the pain scales to directing a multimodal approach to the management of chronic OA pain. Changes and additions are in red font.
  • The limitation of the cross-sectional design is acknowledged, but potential selection or measurement biases are not discussed.
    • Authors: Thank you for your comment. We have mentioned potential selection bias in the cohort given the broad inclusion criteria of our study but still emphasizing how these could provide generalizable results in regard to the associations found as we believe our sample reflects the general KOA population as it is in the Brazilian cohort. Changes and additions are in red font.
  • There is an interesting work that develops to identify the differences in blood investigations between total hip and total knee replacement. I recommend the authors to discuss it: 10.1097/TGR.0000000000000337
    • Authors: Thank you for your comment. We have added a couple of sentences indicating there might be more biological variables associated with these scales and pain perception, further expanding the hypothesis creation from our results. Changes and additions are in red font.
  1. Conclusions
  • The conclusions are sound but could better emphasize practical applications and future research directions.
    • Authors: Thank you for your comment. We have further expanded the applicability of these findings to the clinical setting and correlated it with possible treatment protocols that could be tested in future research regarding KOA chronic pain management. Changes and additions are in red.

Final Recommendations

  • Strengthen the methodological description, particularly regarding selection criteria and variable control.
  • Improve clarity in presenting statistical results and their clinical interpretation.
  • Expand on the therapeutic implications of the findings.
  • Consider additional analyses to reinforce the validity of predictive models.

Overall, the manuscript presents a well-designed study with relevant findings, but certain improvements in methodology and discussion could enhance its impact in the scientific literature.

Best regards,

Authors: Thank you for all your insightful and constructive comments. We have made the appropriate changes and additions to our paper, based on your suggestions. We hope these modifications are agreeable with your comments and they have definitely strengthened the quality of our manuscript.

Reviewer 2 Report

Comments and Suggestions for Authors

General characteristics and evaluation of the reviewed article:

The article addresses the important issue of pain mechanisms associated with knee osteoarthritis (KOA) and their relationship to compensatory neural mechanisms. A multivariate model of pain intensity prediction using psychological, neurophysiological (TMS, EEG) and clinical data is presented. The results indicate the different nature of pain mechanisms in relation to activity (WOMAC) and non-specific pain (VAS), making a valuable contribution to a better understanding of chronic pain in KOA.

The article is well written and based on sound research methods. However, several aspects need to be improved to increase clarity, quality of interpretation of results and potential application in clinical practice. I provide detailed comments below.

Minor comments:

The introduction is far too short and needs to be expanded especially on the epidemiology of OA with the addition of recent references.

Expanding the discussion of osteoarthritis in the introduction could significantly enhance the introduction by highlighting the importance and relevance of this condition. The prevalence of osteoarthritis is influenced by various factors, including occupational activities, sports participation, musculoskeletal injuries, obesity, and gender. Incorporating detailed information about these factors, supported by relevant literature, would provide a robust foundation for the topic. The following references are recommended for inclusion in this section:

https://doi.org/10.3390/healthcare12161648

DOI: 10.1056/NEJMcp1903768

The article is cross-sectional, which means it is difficult to determine whether reduced cortical inhibition and other neurophysiological variables are the cause or effect of chronic pain. My suggestion is to consider conducting longitudinal studies which could strengthen the conclusions. This will help confirm the directionality of the relationship between neurophysiological variables and KOA pain, the authors should consider conducting longitudinal studies. Better balancing the gender of the participants and analyzing gender differences in pain mechanisms could enrich the interpretation of the results. Please discuss this aspect in the discussion highlighting plans for further work.

The majority of participants were female, which may limit the ability to generalize the results to the male population. It would be worthwhile in the future to include more gender representation and to provide analyses stratified by gender. Please describe this limitation in more detail.

The article focuses on comparing the WOMAC and VAS, but does not address alternative pain assessment methods such as the Brief Pain Inventory (BPI) or the McGill Pain Questionnaire. Comparing the effectiveness of different scales could provide valuable information. Adding other pain assessment tools, such as the BPI or McGill Pain Questionnaire, could improve the comparability of results with other studies.

It is unclear whether participants received pain pharmacotherapy or other interventions that could affect neurophysiological outcomes (e.g., EEG, TMS). It would need to be clarified whether pharmacotherapy was controlled for in the analyses. The authors should state whether and which pain medications were used by participants and whether their effects were included in the analyses.

The authors do not analyze differences in pain mechanisms according to the severity of KOA (e.g., Kellgren-Lawrence classification). This could affect the interpretation of the results. Differentiating results according to disease stage could provide valuable clinical information. It would be worthwhile to expand the discussion with plans to relate the results obtained to the degree of damage determined by static imaging studies or a test that allows assessment of the moving joint, e.g., vibroarthrography. The authors will find more detailed information in the papers: Multi-Scale Analysis of Knee Joint Acoustic Signals for Cartilage Degeneration Assessment; Application of Recurrence Quantification Analysis in the detection of osteoarthritis of the knee with the use of vibroarthrography; Application of EEMD-DFA algorithms and ann classification for detection of knee osteoarthritis using vibroarthrography;

Author Response

General characteristics and evaluation of the reviewed article:

The article addresses the important issue of pain mechanisms associated with knee osteoarthritis (KOA) and their relationship to compensatory neural mechanisms. A multivariate model of pain intensity prediction using psychological, neurophysiological (TMS, EEG) and clinical data is presented. The results indicate the different nature of pain mechanisms in relation to activity (WOMAC) and non-specific pain (VAS), making a valuable contribution to a better understanding of chronic pain in KOA.

The article is well written and based on sound research methods. However, several aspects need to be improved to increase clarity, quality of interpretation of results and potential application in clinical practice. I provide detailed comments below.

Minor comments:

The introduction is far too short and needs to be expanded especially on the epidemiology of OA with the addition of recent references.

  • Authors: Thank you for the feedback, we have updated the introduction.

Expanding the discussion of osteoarthritis in the introduction could significantly enhance the introduction by highlighting the importance and relevance of this condition. The prevalence of osteoarthritis is influenced by various factors, including occupational activities, sports participation, musculoskeletal injuries, obesity, and gender. Incorporating detailed information about these factors, supported by relevant literature, would provide a robust foundation for the topic. The following references are recommended for inclusion in this section:

https://doi.org/10.3390/healthcare12161648

DOI: 10.1056/NEJMcp1903768

  • Authors: Thank you for the feedback, we have done the expansion

The article is cross-sectional, which means it is difficult to determine whether reduced cortical inhibition and other neurophysiological variables are the cause or effect of chronic pain. My suggestion is to consider conducting longitudinal studies which could strengthen the conclusions. This will help confirm the directionality of the relationship between neurophysiological variables and KOA pain, the authors should consider conducting longitudinal studies. Better balancing the gender of the participants and analyzing gender differences in pain mechanisms could enrich the interpretation of the results. Please discuss this aspect in the discussion highlighting plans for further work.

  • Authors: This is discussed in the paper limitations section. We just adjusted the section to be clearer, thank you.

The majority of participants were female, which may limit the ability to generalize the results to the male population. It would be worthwhile in the future to include more gender representation and to provide analyses stratified by gender. Please describe this limitation in more detail.

  • Answer: As stated in the introduction and the discussion, female sex is a risk factor for knee OA, therefore, the OA population are composed most by women. Thus, we think that our sample is representative, and this characteristic is not a limitation of our study.

The article focuses on comparing the WOMAC and VAS, but does not address alternative pain assessment methods such as the Brief Pain Inventory (BPI) or the McGill Pain Questionnaire. Comparing the effectiveness of different scales could provide valuable information. Adding other pain assessment tools, such as the BPI or McGill Pain Questionnaire, could improve the comparability of results with other studies.

  • Authors: Thank you for the suggestion and we agree with this comment. In future studies we will try to include these variables.

It is unclear whether participants received pain pharmacotherapy or other interventions that could affect neurophysiological outcomes (e.g., EEG, TMS). It would need to be clarified whether pharmacotherapy was controlled for in the analyses. The authors should state whether and which pain medications were used by participants and whether their effects were included in the analyses.

  • Answer: Unfortunately, the pain medications in use was not a variable in our study. In any case, thank you for the insight, since it can help us to improve future studies in the field.

The authors do not analyze differences in pain mechanisms according to the severity of KOA (e.g., Kellgren-Lawrence classification). This could affect the interpretation of the results. Differentiating results according to disease stage could provide valuable clinical information. It would be worthwhile to expand the discussion with plans to relate the results obtained to the degree of damage determined by static imaging studies or a test that allows assessment of the moving joint, e.g., vibroarthrography. The authors will find more detailed information in the papers: Multi-Scale Analysis of Knee Joint Acoustic Signals for Cartilage Degeneration Assessment; Application of Recurrence Quantification Analysis in the detection of osteoarthritis of the knee with the use of vibroarthrography; Application of EEMD-DFA algorithms and ann classification for detection of knee osteoarthritis using vibroarthrography;

  • Authors: Thank you for the insights. We expanded the discussion with this information. But, as we find in our study and in previous studies the severity of OA is not associated with the severity of chronic pain, therefore the severity of the tissue damage is not the priority of our field of study.

Authors: Thank you for all your insightful and constructive comments. We have made the appropriate changes and additions to our paper, based on your suggestions. We hope these modifications are agreeable with your comments and they have definitely strengthened the quality of our manuscript.

Round 2

Reviewer 1 Report

Comments and Suggestions for Authors

Dear Authors,

I would like to sincerely congratulate you on the quality of the second version of your manuscript entitled "Maladaptive compensatory neural mechanisms associated with activity-related osteoarthritis pain: dissociation of psychological and activity-related neural mechanisms of WOMAC pain and VAS pain."

The revision reflects a rigorous effort and a significant improvement in methodological clarity, depth of analysis, and interpretation of results. In particular, the integration of neurophysiological variables within multivariate pain prediction models represents a valuable contribution to the field of osteoarthritic pain research.

For these reasons, I consider the manuscript ready for publication in the Journal of Clinical Medicine and recommend its final acceptance.

Congratulations once again on this excellent work.

Best regards,